# Systemic Sclerosis and Idiopathic Portal Hypertension: Report of a Case and Review of the Literature

**DOI:** 10.3390/life12111781

**Published:** 2022-11-04

**Authors:** Michele Colaci, Maria Letizia Aprile, Domenico Sambataro, Gianluca Sambataro, Lorenzo Malatino

**Affiliations:** Rheumatology Clinic, Internal Medicine Unit, Azienda Ospedaliera per l’Emergenza Cannizzaro, University of Catania, 95131 Catania, Italy

**Keywords:** systemic sclerosis, scleroderma, idiopathic portal hypertension

## Abstract

**Simple Summary:**

The presence of liver involvement in systemic sclerosis (SSc) is considered atypical, besides the eventual coexistence of other autoimmune hepatic disorders. However, the occurrence of syndromes called idiopathic portal hypertension (IPH) and regenerative nodular hyperplasia (RNH) have been anecdotally reported in the literature. We described a case of SSc complicated by IPH and we reviewed the literature on the topic. No specific SSc pattern linked to IPH emerged, even though the reports often described IPH in patients with limited skin subset SSc. Coexistence of prothrombotic states and overlap with other hepatic diseases could facilitate IPH onset. In spite of IPH being a rare condition, the rheumatologists should consider IPH as a possible hepatic complication in SSc patients.

**Abstract:**

The presence of liver involvement in systemic sclerosis (SSc) is considered atypical, besides the possible coexistence of other autoimmune hepatic disorders. However, the occurrence of portal hypertension and, more specifically, of the syndromes called idiopathic portal hypertension (IPH) and regenerative nodular hyperplasia (RNH) have been anecdotally reported in the literature for SSc patients. We described a case of SSc woman complicated by IPH; moreover, we reviewed the literature on the topic. A 61-year-old female SSc patient was admitted to our hospital because of the onset of ascites. SSc, as a limited skin subset of disease with anticentromere antibodies, was diagnosed 11 years previously, with no significant visceral involvement. We excluded possible causes of portal hypertension, namely chronic infections, autoimmune hepatic diseases, neoplasia, thrombosis of portal vein, and Budd–Chiari syndrome. Finally, IPH was diagnosed. A review of the literature identified a number of case reports or case series that described IPH in the course of SSc. No specific SSc pattern linked to IPH emerged, even though reports from the literature often described the limited skin subset. Coexistence of prothrombotic states and overlap with other hepatic diseases could facilitate IPH onset. Besides being a rare condition, the onset of IPH in SSc patients is an occurrence that should be taken into account.

## 1. Introduction

Systemic sclerosis (SSc) is an autoimmune connective tissue disease characterized by diffuse vasculopathy and fibrosis of skin and internal organs. Endothelial activation leads to augmented permeability, expression of adhesion molecules, and the secretion of vasomodulators that induce vasoconstriction and platelet aggregation [1]. Furthermore, activated myofibroblasts overproduce disarranged extracellular matrix and collagen, leading to fibrosis of tissues and organs. Widespread microangiopathy extensively alter the capillary architecture, thereby reducing the density of capillary loops [2]. Microvascular derangement is well observable by means of nailfold capillaroscopy that may be considered a reliable tool to estimate the systemic vascular disease [3].

In clinical practice, the liver and its microvascular structure are not included among the targets of SSc. Unlike lungs, heart, kidneys and the digestive tract, the liver is usually spared by fibrosis, therefore cirrhosis and liver dysfunction are not reported as typical SSc features [4,5]. Nonetheless, SSc patients can infrequently present an autoimmune involvement of the liver, mainly consisting in primary biliary cholangitis (PBC), as an overlap autoimmune disorder [6]. In a Spanish registry recruiting 1572 SSc patients, 118 (7.5%) cases of hepato-biliary disorders were recorded: PBC was the main cause in over the half of the cases, while autoimmune hepatitis represented 16% of this group; finally, only seven patients showed hepato-biliary abnormalities attributed to SSc itself [6]. 

In this paper, we focused on idiopathic portal hypertension (IPH). We presented the case of a SSc patient who received this diagnosis; subsequently, we conducted a review of the literature on the association between SSc and IPH.

## 2. Case Report

In January 2018, a 61-year-old woman was admitted to our Internal Medicine Unit because of the onset of ascites of unknown origin. She had been affected by SSc since 2007. The disease was characterized by limited skin subset, presence of diffuse telangiectasias on the face, absence of digital ulcers or calcinosis, typical scleroderma esophagopathy, no significant pulmonary, cardiac or renal involvement, and detection of antinuclear antibodies 1:1280 with anti-centromere pattern. Overall, the patient was sarcopenic (body weight was 46.6 kg, BMI 18 Kg/m^2^). 

The diagnostic work-up included several exams in order to find the aetiology of ascites. Computed tomography (CT) of the abdomen excluded gynaecological diseases, i.e., cancer with peritoneal carcinosis, firstly suspected since the detection of increased serum levels of Ca125 (163 UI/mL). By contrast, the CT showed a slight reduction of liver volume, without focal lesions and increased diameter of the portal vein (17 mm). Paraumbilical collateral venous circulation was evidenced, as well as an augmented volume of the spleen. Pleural-pericardial effusion was also present. The tests for anti-mitochondrial, anti-smooth muscle cell and anti-LKM antibodies were negative. Moreover, the measurement of IgG subclasses, ferritin and ceruloplasmin did not show alterations. Liver function tests were almost in range, as well as coagulation proteins, transaminases, and cholestasis indices. HBV, HCV and HIV infections, as well as alcohol or drug abuses, were firmly excluded. Hence, we decided to program a liver needle biopsy, after improvement of ascites with the use of diuretics, in order to verify the hypothesis of overlap syndrome with an autoimmune hepatitis.

After a weight loss of 5 kg (ascites almost cleared up) and a general improvement, the patient underwent liver biopsy that showed normal architecture, no necrosis or inflammation, mild fibrosis of the portal spaces, and vacuolar degeneration of several hepatocytes. 

In parallel, the patient underwent oesophagogastroduodenoscopy that did not show varices, but only erosive oesophagitis with a small hiatal hernia.

The follow-up was characterized by a stable and fair general condition one year later, with the administration of potassium kanrenoate 50 mg daily. Subsequently, the patient dropped out of our follow-up because of admission to another hospital for sepsis causing death. 

## 3. Review of the Literature

IPH is a rare syndrome characterized by the development of intrahepatic portal hypertension in absence of cirrhosis. An increased interest in this condition has developed since its increasing diagnosis in patients with autoimmune disorders, as well as haematological, neoplastic, and infectious diseases [7,8,9,10]. Regarding the pathophysiology of IPH, it was hypothesised that an initial injury to the intrahepatic vascular bed would cause an increase of intrahepatic resistance to portal blood flow [8]. The etiology of IPH remains unknown and many theories have been proposed [11]. In particular, the infection theory suggests that repeated infection of intestinal bacteria, such as Escherichia coli, might cause septic embolizations towards the intrahepatic portal veins, leading to progressive small vessel occlusions. Even exposure to toxics, such as arsenic or medications, has been reported as causing IPH [7,11]. Moreover, thrombophilia was indicated as another possible cause of IPH, suggested by the high prevalence of prothrombotic disorders and high incidence of portal vein thrombosis in IPH patients. Of interest, the autoimmune origin of IPH has been suggested due to a female preponderance and overlap with other autoimmune diseases such as systemic lupus erythematous or SSc itself [7,8,9,10,11]. To date, the actual role of each these etiologic factors remains speculative. 

In the scientific literature, IPH is usually associated with regenerative nodular hyperplasia (RNH), also named porto-sinusoidal vascular disease or, less frequently, non-cirrhotic portal hypertension, hepato-portal sclerosis, or incomplete septal cirrhosis, emphasizing the histological alteration of the hepatic parenchyma. This nomenclature heterogeneity contributed to leave this condition largely under-recognized. The term porto-sinusoidal vascular disease, proposed by the European Association for the Vascular Liver Disease (VALDIG), aimed to underline that this pathological entity may be identifiable, satisfying the diagnostic criteria and not only by exclusion, as suggested by the term “idiopathic” [12].

As described by Wanless et al. [13] in a report of 64 RNH cases out of 2500 autopsies, the development of NRH in IPH is caused by the narrowing and obstruction of the portal vein branch that leads to atrophy of the lobules with poor circulation, and the compensatory hyperplasia of the other lobules that maintain the blood flow.

The diagnosis is based on the presence of the signs of portal hypertension (namely, the presence of gastro-oesophageal varices, ascites, splenomegaly), the absence of cirrhosis or advanced fibrosis due to known chronic liver diseases that should be appropriately excluded, and the absence of thrombosis of the hepatic or of the portal veins [7]. Unfortunately, there are no specific diagnostic tests for IPH, but an increasing knowledge of the disease has permitted to propose diagnostic criteria [8]. In particular, IPH diagnosis may be based on the presence of clinical, imaging, and histopathological features. Clinically, the presence of gastro-oesophageal varices or other porto-systemic collaterals and the exclusion of chronic liver diseases causing cirrhosis are the major criteria. Furthermore, the presence of ascites, thrombocytopenia, splenomegaly or increased liver enzymes are minor criteria [8]. Imaging techniques should evidence the absence of portal thrombosis, normal or slightly elevated (<10 mmHg) hepatic venous pressure gradient, normal or slightly increase of liver stiffness at elastography, while CT/MRI should demonstrate sudden narrowing of second-degree intrahepatic portal vein branches and benign hypervascular nodules [8]. Finally, liver biopsy should exclude cirrhosis; in particular, nodular regenerative hyperplasia, incomplete septal cirrhosis, and obstructive portal venopathy are major criteria, while focal sinusoidal dilation, perisinusoidal fibrosis, lobular architectural disruption and other alterations compatible with portal hypertension are minor criteria [8]. 

Liver biopsy is mandatory for patients with overt signs of portal hypertension in order to confirm or exclude the presence of cirrhosis. Furthermore, liver biopsy is necessary in order to demonstrate the histological signs of RNH. However, the histological exam may have some limitations because the histopathological alterations are not always present in all IPH patients nor are they widely distributed in the liver, thus reducing the possibility of reaching diagnosis. This makes it necessary to obtain an adequate liver specimen including the highest possible number of portal tracts, in order to increase the possibility of finding the typical RNH lesions [14].

IPH and RNH have been occasionally described in SSc patients [6,15,16,17,18,19,20,21,22,23,24,25,26,27,28,29,30,31,32,33,34,35,36,37,38,39,40,41,42,43]. (Table 1 and Table 2). Single cases or small case series have reported SSc patients who presented also IPH during follow-ups. However, nodular hyperplasia and hyperplastic changes were observed in only one third of IPH associated with autoimmune diseases [26].

Considering the 41 subjects summarised in Table 1 and Table 2, we observed a clear prevalence of women, as expected for SSc. Mean age was 55 years (range 33–82). Japanese patients were the most frequent cases reported, even though research using the term RNH revealed that several cases can be found also in western countries. On the basis of the available data, those with black ethnicity seemed to be spared. Regarding SSc subsets, the limited skin subset of SSc, formerly referred to as CREST (acronym of: calcinosis, Raynaud, esophagopathy, sclerodactyly, telangiectasias) tend to be prevalent, such as with our case. Clinically, complications of portal hypertension, i.e., esophageal varices and their bleeding or ascites, determine the prognosis. 

Clinical histories of the SSc patients described in the literature showed that in several cases, other pathologic conditions were reported to overlap. In a few cases, Sjogren’s syndrome was reported, in others the presence of coagulation disorders emphasised the possible role of a prothrombotic state in IPH pathophysiology. Finally, studies showing an overlap with PBC were excluded from the present review to better focus attention on IPH/RNH.

## 4. Discussion

Hepatopathy is not considered a typical organ involvement in the course of SSc [4,5], even though a number of reports in the literature [7,8,9,10,11,12,13,14,15,16,17,18,19,20,21,22,23,24,25,26,27,28,29,30,31,32,33,34,35,36,37,38,39,40,41,42,43] and the findings of the Spanish RESCLE Registry [6] emphasised the possibility of hepatic disorders in SSc patients. Besides the overlap with autoimmune liver disorders, such as PBC or autoimmune hepatitis, we focused on the pathologic syndrome named IPH. This latter syndrome represented the final diagnosis of our SSc patient. In fact, she showed overt signs of portal hypertension, without the features of other hepatic diseases of infectious, autoimmune or neoplastic origin. However, our patient did not show the histologic and instrumental multinodular pattern that is typical of RNH, possibly because of the early onset of the disease besides the abrupt development of portal hypertension. Indeed, IPH is characterized by heterogeneity in its histological presentation [10] and RNH is not constantly reported in IPH patients. In general, the histological pattern of IPH is the presence of thickening, narrowing and obliteration of small and medium-sized portal veins. Subsequently, portal tracts tend to approximate each other, thereby producing parenchymal atrophy [10], with no or minimal septal fibrosis. 

One of the main hallmarks of SSc is diffuse vasculopathy with endothelial dysfunction [1]. Moreover, it has been demonstrated that cells in endothelial-to-mesenchymal transition in the vessels of SSc patients could link endothelial dysfunction to the development of tissue fibrosis [44]. We might speculate that this condition may contribute to IPH pathophysiology, providing in turn a favouring background. It was assumed that, after a hypothetical trigger to the intrahepatic microcirculation, the obliterative portal damage due to hepatoportal sclerosis led to the increase of vascular resistance, hence to portal hypertension [8]. Of interest, anti-endothelial antibodies (AECA), expressed in some patients with IPH damaged endothelial cells of portal vessels and produce dense deposits of elastic fibres around the peripheral ramifications of the portal vein [45]. AECA were also found in a significant percentage of SSc patients, directly correlated with vascular injury and endothelial damage [46], by means of antibody-dependent cellular apoptosis that stimulated the microvasculature to release pro-inflammatory and pro-fibrotic cytokines [47].

Despite IPH pathophysiology being strictly linked with endothelial damage and dysfunction, the presence of IPH in SSc is a rare occurrence. Therefore, an initial injury to portal microvasculature should be postulated as a necessary trigger for IPH. It was proposed that the repeated microthrombosis in the small or medium portal vein branches was a plausible trigger [48]. Indeed, this hypothesis could be matched with the observation of IPH in prothrombotic conditions, such as that observed in at least a subset of SSc patients, mainly when associated with other coagulation disorders, i.e., the presence of anti-phospholipid antibodies. Other potential triggers supporting IPH, even in SSc patients, could be the exposure to toxics, infection by hepatotropic viruses or an overlap with autoimmune hepatic diseases [10]. 

To date, IPH in SSc represents a condition reported anecdotally. However, the possible association between SSc and other autoimmune disorders prompts the inclusion of a liver examination in the periodic work-up of SSc patients, by means of ultrasound imaging of the abdomen. However, the therapeutic approach to SSc patients with IPH remains to be understood. In our case, we prescribed mycophenolate mofetil as an immunosuppressive agent able to control SSc-related inflammation and fibrosis. Indeed, the rationale for this therapeutic strategy was that IPH was a consequence of disease progression. To the best of our knowledge, no significant data addressing the treatment of IPH complicating SSc can be found in the literature. In a few cases, the successful administration of steroids has been reported [27,49], suggesting an autoimmune pathogenesis of IPH. However, this issue remains to be further investigated.

## 5. Conclusions

In conclusion, we presented the case of a SSc patient who developed portal hypertension diagnosed as IPH. The review of the literature found other similar cases, even considering the reports in which IPH was termed RNH. Even though it is a rare condition, liver disease and portal hypertension in the course of SSc should be taken into account.

## Figures and Tables

**Table 1 life-12-01781-t001:** SSc patients with IPH described in literature. Legend: skin subset (L = limited; D = diffuse); serology (ANA = antinuclear antibodies, ACA = anticentromere antibodies). NA = data not available.

First Author/Year	Age/Sex	Country	Skin Subset	Serology	Interventions for IPH	Notes
Morris/1972 [15]	53F	UK	L	ANA	Porto-caval anastomosis	
Umeyama/1982 [16]	41F	Japan	D	ANA	NA	Pancytopenia, hypergammaglobulinemia
Tajima/1985 [17]	45F	Japan	D	ND	NA	
Nakanuma/1991 [18]	45F	Japan	D	ND	NA	
Sanchez/1997 [19]	54F	Spain	L	ND	propanolol and isosorbide 5-mononitrate prophylaxis, then portacaval shunt	Protein S deficiency
Asai/1998 [20]	58F	Japan	L	ACA	NA	
Watanabe/1999 [21]	44F	Japan	D	ND	Splenectomy	Sjogren’s syndrome overlap
Pérez Garcìa/2002 [22]	59M	Spain	D	Neg.	Propranolol prophylaxis	
Tsuneyama/2002 [23]	57F; 60F	Japan	D	ACA; Scl70	None	2 case: non cirrhotic HCV+
Ishii/2003 [24]	53F	Japan	D	Scl70	None	Heterozigosis for factor V Leiden mutation
Moschos/2005 [25]	82M	UK	ND	ND	Propranolol prophylaxis, spironolactone 100 mg/day, EV ligation	
Kogawa/2005 [26]	72F	Japan	L	ACA, Scl70 SSA, M2	None	Sjogren’s syndrome overlap
Takagi/2006 [27]	62F	Japan	L	U1-RNP	prednisolone and EV ligation	
Ortega Espinosa/2014 [28]	38F	Brazil	L	ACA	EV ligation, propranolol prophylaxis	
Yamamoto/2021 [29]	76F	Japan	L	ACA, ssDNA	Furosemide, spironolactone, tolvaptan	

**Table 2 life-12-01781-t002:** SSc patients with NRH described in literature [6,30,31,32,33,34,35,36,37,38,39,40,41,42,43]. Legend: skin subset (L = limited; D = diffuse; ss = sine scleroderma); serology (ANA = antinuclear antibodies, ACA = anticentromere antibodies). NA = data not available. CREST = acronym of: calcinosis, Raynaud, esophagopathy, sclerodactyly, telangiectasias; a subset of SSc. PAH = pulmonary arterial hypertension. Pt(s) = patient(s).

First Author/Year	Age/Sex	Country	Skin Subset	Serology	Interventions for IPH	Notes
Lurie/1973	1 pt	Israel	L	ND	NA	CREST
Russell/1983	1 pt	Canada	ND	ND	NA	
Friguet/1984	1 pt	France	ND	ND	NA	
Cadranel/1987	1 pt	France	ND	ND	NA	CREST
García Díaz/1989	52F	Spain	L	ND	NA	CREST
Wanless/1990	5 pts	Canada	ND	ND	NA	2 CREST
Perez Ruiz/1991	69F	Spain	L	ANA	NA	
Kaburaki/1996	33F	Japan	ND	U1RNP	Diuretics	Sjogren’s syndrome overlap
Agard/2000	62F	France	ND	ACA	-	CREST, small B cells lymphoma
Matsumoto/2000	F	Japan	ND	ND	NA	
Morris/2010	1 pt	UK	ND	ND	NA	
Mendel/2011	57F	Canada	L	ND	Diuretics	CREST with PAH
Kamel/2016	1 pt	USA	ss	ACA, DNA	NA	PAH
Graf/2018	4 pts	Switzerland	ND	ND	NA	
Abrams/2018	59M	USA	L	ACA	TIPS	CREST
Marì-Alfonso/2018	3 pts	Spain	L	ND	NA	

## Data Availability

Not applicable.

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
