# Peer review of "Systemic Sclerosis and Idiopathic Portal Hypertension: Report of a Case and Review of the Literature"

_life, 2022, doi:10.3390/life12111781_

Round 1
Reviewer 1 Report
I read your article with interest. It is generally well written and addresses a very rare phenomenon in SSc. I have a few comments:
Language: a thorough proofreading by an English native is necessary.
61: Mass without height does not define cachectic state. In fact, a woman of low height 155 cm at 47 kg is having normal BMI.
69: "research" is not an appropriate word here. "Tests for" or "serological analyses of" would be better.
72: instead of "regards" you probably mean "as well as"
73: Infections from is not grammatically correct
77: you cannot "solve" ascites"
79: I am not an expert in hepatology but isn't fibrosis of portal spaces an early stage of liver fibrosis before cirrhosis? please explain briefly in the discussion (lines: 116-134) how it relates to the described case...
Lines 9-11 & 103 - According to the literature "Nodular Regenerative Hyperplasia is one of the causes of noncirrhotic portal hypertension" - (see publications below). Thus, you cannot use these terms interchangeably, because they mean something slightly different than idiopathic portal hypertension. Please rewrite these paragraphs because liver parenchymal changes lead to IPH but they are not synonyms.
In the text you also use NRH and sometimes RNH - please change it to NRH.
World J Gastroenterol. 2011 Mar 21; 17(11): 1400–1409.
Published online 2011 Mar 21. doi: 10.3748/wjg.v17.i11.1400
J Investig Med High Impact Case Rep
. 2021 Jan-Dec;9:23247096211044617. doi: 10.1177/23247096211044617.
Line 103 - please refrain from using "Western scientific literature" - there are no borders in science and it may be insulting to many scientists all around the world. Is MDPI journal publishing Italian scientist Western? That distinction makes no sense (except social sciences and history), even in the geographical sense.
152: CREST is no longer a recommended term for limited cuteaneous Systemic Sclerosis (lcSSc) as adopted in ACR/EULAR criteria. Please use lcSSc throughout the text.
Classification Criteria for Systemic Sclerosis: An ACR-EULAR Collaborative Initiative
Arthritis Rheum. 2013 Nov; 65(11): 2737–2747.
Wollheim FA. Classification of systemic sclerosis: visions and reality. Rheumatology (Oxford) 2005;44:1212–6
218: "develop" => developed (tense)
Author Response
I read your article with interest. It is generally well written and addresses a very rare phenomenon in SSc. I have a few comments: Language: a thorough proofreading by an English native is necessary.
The English of the paper has been revised. Please, find the changes.
61: Mass without height does not define cachectic state. In fact, a woman of low height 155 cm at 47 kg is having normal BMI.
Yes, the reviewer is right. We have added the BMI and have changed cachectic to sarcopenic.
69: "research" is not an appropriate word here. "Tests for" or "serological analyses of" would be better.
Made the change.
72: instead of "regards" you probably mean "as well as”
Error fixed.
73: Infections from is not grammatically correct
Error fixed.
77: you cannot "solve" ascites”
We have changed to the term “clear up”.
79: I am not an expert in hepatology but isn't fibrosis of portal spaces an early stage of liver fibrosis before cirrhosis? please explain briefly in the discussion (lines: 116-134) how it relates to the described case…
At the end of the first paragraph of the Discussion we have written: “In general, the histological pattern of IPH is the presence of thickening, narrowing and obliteration of small and medium-sized portal veins. Subsequently, portal tracts tend to approximate each other, so producing parenchymal atrophy [10].” This is different form the presence of fibrosis of portal spaces (up to cirrhosis), because fibrosis is minimal or absent. We added “…with no or minimal septal fibrosis”.
Lines 9-11 & 103 - According to the literature "Nodular Regenerative Hyperplasia is one of the causes of noncirrhotic portal hypertension" - (see publications below). Thus, you cannot use these terms interchangeably, because they mean something slightly different than idiopathic portal hypertension. Please rewrite these paragraphs because liver parenchymal changes lead to IPH but they are not synonyms.
Of course, we agree with the reviewer. Thus, we tried to change the text in order to eliminate misunderstandings.
In the text you also use NRH and sometimes RNH - please change it to NRH.
Error fixed.
Line 103 - please refrain from using "Western scientific literature" - there are no borders in science and it may be insulting to many scientists all around the world. Is MDPI journal publishing Italian scientist Western? That distinction makes no sense (except social sciences and history), even in the geographical sense.
We made the change.
152: CREST is no longer a recommended term for limited cuteaneous Systemic Sclerosis (lcSSc) as adopted in ACR/EULAR criteria. Please use lcSSc throughout the text.
We know that the acronym “CREST” is outdated and no longer used. We used it because it referred to the clinical cases described in the table. However, with the reviewer right, we have corrected this expression.
218: "develop" => developed (tense)
Error fixed.
Finally, we would thank the reviewer for the precious comments that permitted the improvement of our paper.
Reviewer 2 Report
 Colaci et al provide a summary of cases, a summary of the literature, and a discussion of a rare presentatoon, IPH associated with SSc. PBC and AIH are commonly experienced as liver diseases associated with SSc, but IPH, as described in the text, is difficult to detect as an advanced disease without PBC or AIH. Moreover, as in the present case, when the skin sclerosis is not severe and there is no interstitial pneumonia or ulceration, the patient is often only observed without meaningful treatment, and as a result, the condition progresses. This persistence of microinflammation leading to fibrosis and pathology is a major problem in the treatment of scleroderma. In this case, MMF was used after IPH was identified, but we cannot help but wish that we had been able to treat the patient earlier before this occurred. There were no particular concerns regarding the wording or descriptions in the text.
 I hope that detailed analysis will reveal which patients have risk factors that predispose them to IPH, so that they can be treated prophylactically with immunosuppressive drugs such as MMF. I believe this is an excellent case presentation and summary of the literature that raises such issues and future perspectives in SSc treatment.
Author Response
We would thank the reviewer for the appreciation of the article. The aim of the paper has been clearly understood, considering the number of open questions and unmet needs that a case like the present can raise. Again, thanks.